# Annular Tautomerism of 3(5)-Disubstituted-1*H*-pyrazoles with Ester and Amide Groups

**DOI:** 10.3390/molecules24142632

**Published:** 2019-07-19

**Authors:** Anna Kusakiewicz-Dawid, Monika Porada, Błażej Dziuk, Dawid Siodłak

**Affiliations:** Faculty of Chemistry, University of Opole, Oleska 48, 45-052 Opole, Poland

**Keywords:** pyrazole, tautomer, conformation, X-ray, NOE, FT-IR, NICS, DFT

## Abstract

A series of disubstituted 1*H*-pyrazoles with methyl (**1**), amino (**2**), and nitro (**3**) groups, as well as ester (**a**) or amide (**b**) groups in positions 3 and 5 was synthesized, and annular tautomerism was investigated using X-ray, theoretical calculations, NMR, and FT-IR methods. The X-ray experiment in the crystal state showed for the compounds with methyl (**1a**, **1b**) and amino (**2b**) groups the tautomer with ester or amide groups at position 3 (tautomer 3), but for those with a nitro group (**3b**, **4**), tautomer 5. Similar results were obtained in solution by NMR NOE experiments in CDCl_3_, DMSO-*d*_6_, and CD_3_OD solvents. However, tautomer equilibrium was observed for **2b** in DMSO. The FT-IR spectra in chloroform and acetonitrile showed equilibria, which can be ascribed to conformational changes of the cis/trans arrangement of the ester/amide group and pyrazole ring. Theoretical analysis using the M06-2X/6-311++G(d,p) method (in vacuo, chloroform, acetonitrile, and water) and measurement of aromaticity (NICS) showed dependence on internal hydrogen bonds, the influence of the environment, and the effect of the substituent. These factors, pyrazole aromaticity and intra- and inter-molecular interactions, seem to have a considerable influence on the choice of tautomer.

## 1. Introduction

Pyrazole is a five-membered aromatic heterocycle with two adjacent nitrogen atoms, classified as azole. Pyrazole derivatives have broad application, from engineering and polymers [1,2,3,4,5,6,7,8], to biological activity [9]; thus, new synthetic strategies are constantly developed [10,11,12,13,14,15,16,17]. Among them, there are 1*H*-pyrazoles with substituents at positions 3 and 5, which reveal interesting applications [18,19,20,21,22,23,24,25], and some of them are depicted in Figure 1. The substitution at positions 3 and 5 by amide, ester, or generally a carbonyl linkage makes a convenient structural building block of potentially broad usage. For example, a non-natural 1*H*-pyrazole amino acid was investigated as a building block of short peptides designed to act as β-sheet ligands, which prevent protein aggregation. The distinguishing properties of the 1*H*-pyrazole residue result from a specific hydrogen bonding pattern, in which hydrogen bond donors and acceptors can be aligned on one side of the residue [26,27,28]. To get more detailed information about these properties, the conformational properties of the 3-amino-1*H*-pyrazole-5-carboxylic acid residue were investigated. It was found that the observed conformational preferences were clearly related to the presence of intramolecular interactions formed within the studied residue [29]. This shows that the *N*-unsubstituted 1*H*-pyrazole moiety is able to create a specific type of hydrogen interaction and, then, association, which can be crucial for biological properties, or even broader for materials science. However, these properties strictly depend on the position of the hydrogen atom. Therefore, annular tautomerism of the 1*H*-pyrazoles is the subject of interest.

The majority of the literature data is concerned with various 3(5)-substituted pyrazole compounds, investigated mostly using X-ray structure analysis and NMR spectroscopy, such as 3(5)-phenyl- and 5(3)-methyl-3(5)-phenylpyrazole [30], single aryl substituents [31], 3(5)-amino-5(3)-arylpyrazoles [32], 3-nitropyrazole [33], 3(5)-aminopyrazoles [34], 1*H*-pyrazole-3-(*N*-tertbutyl)-carboxamide [35], or 3(5)-trifluoromethyl-1*H*-pyrazoles [36]. Various monosubstituted pyrazoles were studied using theoretical methods on isolated molecules [37,38]. However, there is still insufficient data concerning 3(5)-substituted pyrazole with an amide or ester linkage.

In this work, we present a study on the annular tautomerism of 3(5)-disubstituted-1*H*-pyrazoles, with an ester/amide bond and methyl, amine, and nitro substituents (Figure 2). Assuming that intra- and inter-molecular interactions can be important factors, not only the position of the hydrogen atom at the pyrazole nitrogen atom, but also the conformation of the carbonyl linkage are considered.

## 2. Results and Discussion

### 2.1. X-Ray Results

Molecular structures of the studied molecules **1a**, **1b**, **2b**, **3b**, and **4** in the crystal state are presented in Figure 3. Molecular interactions are presented in Figure 4. The X-ray experimental details are presented in Table 1 and selected geometric parameters in Appendix A.

In the crystal structure of *N*-methyl 5-methyl-*1H*-pyrazole-3-carboxylamide (**1a**), CH_3_-Pz-CONHCH_3_, there are two types of molecules with slightly different geometries. In both cases, the amide group is at position 3 in relation to the N-H group of the pyrazole ring. Therefore, the tautomer denoted as T3 is present. The torsion angle N2-C1-C5-N3 has a value of −15.42(3)°. This shows that the nitrogen atoms of the amide group and pyrazole ring are at the same side, and the cisoidal conformation (T3c) is obtained. The low value of torsion angle ψ indicates that π-electron conjugation between the pyrazole ring and amide group is possible. Some deformation results from the intermolecular hydrogen bonds. Two molecules of a slightly different geometry create a centrosymmetric dimer stabilized by two N-H…N hydrogen bonds (Appendix A), in which the donor is the N-H group of the amide bond and the acceptor is the nitrogen atom at positon 2 of the pyrazole ring. Additionally, each molecule creates two N-H…O hydrogen bonds involving the C=O group as an acceptor and the N-H group of the pyrazole ring as a donor.

In the crystal structure of ethyl 5-methyl-*1H*-pyrazole-3-carboxylate (**1b**), CH_3_-Pz-COOCH_2_CH_3_, the ester group is also at position 3 in relation to the N-H group of the pyrazole ring, which means that the tautomer T3 is presented. The ester group is almost coplanar with the pyrazole ring (ψ = -5.84°), and thus, the same tautomer and conformer (T3c) are obtained as in the case of the amide analogue. In contrast, the molecules are arranged in a linear association, which is stabilized by the N-H…N hydrogen bonds created between the pyrazole nitrogen atoms. Additionally, the carbonyl oxygen of the ester group creates a short contact with the methyl group at position 2 of the pyrazole ring, whereas the oxygen atom of the alkoxy group is involved in the contact with the pyrazole N-H group.

In the crystal state of methyl 5-amino-*1H*-pyrazole-3-carboxylate (**2b**), H_2_N-Pz-COOCH_3_, three independent molecules are present in the asymmetric unit. Again, the tautomer T3 is adopted. In the case of the ester group, the torsion angle ψ is close to a value of 180°. This means that in contrast to compounds **1a** and **1b**, the transoidal conformation is adopted (T3t), in which the oxygen atom of alkoxy group is in the trans position to the nitrogen atom of the pyrazole ring. Three molecules of slightly different geometry create a trimer stabilized by six hydrogen bonds. Inside the trimer, three N-H…N hydrogen bonds are created by the pyrazole nitrogen atoms, where each pyrazole ring assumes a donor and acceptor function. Three N-H…O hydrogen bonds are created by the NH_2_ groups and carbonyl oxygens of neighboring molecules.

For the molecules of methyl 3-nitro-*1H*-pyrazole-5-carboxylate (**3b**), O_2_N-Pz-COOCH_3_, in the crystal state, the ester group is at positon 5 of the pyrazole ring. Thus, in contrast to previous compounds, the tautomer denoted as T5 is adopted. The ester group is coplanar with the pyrazole ring (ψ = 0°), and the cisoidal conformation (T5c) is present. A linear association is formed, stabilized by the N-H…O hydrogen bonds created by the N-H group of the pyrazole ring and carbonyl ester group. There is also a short C-H…N contact between the C2-H and N2 of the pyrazole ring.

The crystal of methylammonium 3-nitro-*1H*-pyrazole-5-carboxylate (**4**), O_2_N-Pz-COO^–^MeNH_3_^+^, shows that the ester group is at position 5 of the pyrazole ring. Therefore, the tautomer T5 is present. Both the nitro and carboxylate groups are coplanar with the pyrazole ring. In comparison to previous compounds, the association model is more complex. The pyrazole ring creates two hydrogen bonds: as the acceptor in the N^+^-H…N hydrogen bond with the methylammonium cation and as a donor in the N-H…O hydrogen bond with the carboxylate group of another molecule. Additionally, each carboxylate group creates three N^+^-H…O hydrogen bonds with methylammonium cations. The nitro groups are involved in C-H…O contacts with the methyl group of the ammonium cation and the C2-H group of the pyrazole ring of the neighboring molecule.

As can be seen, the studied compounds reveal various association patterns. They create the N-H…N and N-H…O hydrogen bonds, involving both the N-H group of the pyrazole ring and the C=O and N-H groups of ester/amide moieties. Nevertheless, a trend can be seen, for CH_3_-Pz-CONHCH_3_ (**1a**), CH_3_-Pz-COOCH_2_CH_3_ (**1b**), and H_2_N-Pz-COOCH_3_ (**2b**): the amide/ester group is at position 3 of the pyrazole ring (the tautomer T3), whereas in O_2_N-Pz-COOCH_3_ (**3b**) and O_2_N-PzCOO^–^MeNH_3_^+^ (**4**), the ester group is at position 5 (the tautomer T5).

Additional information was obtained by searching the Crystal Structure Database and analysis of the structures containing three and five disubstituted pyrazole rings (Table 2, Appendix A). For the methyl substituent, only the structures of tautomer 3 are present with the value of the torsion angle ψ close to 0° (T3c). In contrast, the structures with the alkyl substituent, but with the fluorine atom at the α carbon, adopt solely tautomer 5. This also occurs for the structures with the nitrile and hydroxyl groups. When the phenyl group is present, all possible structures are adopted, but with preferences towards tautomer 3. It seems that for the choice of tautomer, the type of substituent is important, that is the σ/π-donating/-withdrawing character. When the σ donating methyl group is present, tautomer 3 is adopted. The introduction of the fluorine atom results in the σ-withdrawing character of the substituent; therefore, tautomer 5 is adopted. The nitrile substituent with both the σ and π-withdrawing character results solely in tautomer 5. The data for 3(5)-carboxy-*1H*-pyrazole-5(3)-carboxylate monohydrate shows an interesting balance. The carboxy and carboxylate groups are structurally similar; thus, both tautomers are present, but the negative charge of the carboxylate group results in tautomer T3 being preferred, in which the N-H group of the pyrazole ring is closer to the electron donating substituent (or the pyrazole nitrogen atom with a lone electron pair closer to more electron-withdrawing substituents). This corroborates the result obtained in the case of the structure of **4**. The data in Appendix A show that the amide and ester groups have a similar effect, at least in the solid state.

### 2.2. Theoretical Calculations

In order to get more detailed information, theoretical calculations were performed. Table 3 and Appendix A present the local minima of the studied compounds **1**–**3** in relation to their tautomers (position 3 or 5 of the carbonyl substituent) and rotation of the carbonyl substituent (the torsion angle ψ close to a value of 0° or 180°). Each studied compound has two tautomers (T3 or T5), and each tautomer has two conformations (Tc or Tt, ψ ~ 0° or 180°, respectively). Therefore, for each compound, four local minima (T3c, T3t, T5c, T5t) were analyzed. Thus, the information about the relative energy between tautomers, conformers, and amide/ester substituent can be obtained. The calculations were performed for isolated molecules (in vacuo), as well as mimicking an increase of environment polarity (chloroform, acetonitrile, water).

As can be seen in Table 3, the results obtained for the most polar water-mimicking environment corresponded fairy well with those for the X-ray structures. For CH_3_-Pz-CONHCH_3_ (**1a**), CH_3_-Pz-COOCH_2_CH_3_ (**1b**), and H_2_N-Pz-COOCH_3_ (**2b**) the lowest energy structure is adopted in the crystal state, T3c for **1a** and **1b** and T3t for **2b**. For O_2_N-Pz-COOCH_3_ (**3b**), the second structure in the predicted energy order is adopted (T5c), but the difference in relative energy from the lowest energy structure (T5t) is small.

The influence of the environment is different for each structure. For more clear visualization, the diagrams of the change of relative energy are given in Appendix A. For the pyrazole substituted by the nitro group (**3a** and **3b**), the environment does not seem to have an influence on the choice of tautomer and conformation. The structure T5t is adopted with the ester/amide group in position 5 and the torsion angle ψ ~ 180°. The preferences of the pyrazole ring substituted by the methyl and amide groups (**1a**) also seem to be independent of the polarity of the environment. In this case, the structure T3c is adopted with the amide group in position 3 and the torsion angle ψ ~ 0°. A difference is seen when the ester group is present instead of amide (**1b**). Although the structure in the crystal state and solvent-mimicking environment is the same (T3c), for the isolated molecule **1b**, the structure T5t is predicted. This difference results from the presence of intramolecular hydrogen bonds. As is depicted in Appendix A, in the structure T3c with the amide substituent (**1a**), there are two hydrogen bonds: N-H…N between the amide N-H and pyrazole N2, as well as C-H…O, between the pyrazole C4-H and amide O. The concomitant existence of these forces results in the highest stability of the structure T3c. It should be noticed that for the isolated molecule **1a**, the second in energy order is the structure T5t, which can be explained by the presence of the N-H…O hydrogen bond (between the pyrazole N-H and amide O). The remaining structures, T3t and T5c, undergo unfavorable H…H repulsion, which for T5c is seen by a deviation of the torsion angle ψ. All this together explains why the structure T3c is favorable. However, for the analogue with the ester substituent (**1b**), the structures T5c and T5t have two hydrogen bonds, where the donors are the pyrazole N-H and C-H groups and the acceptors are two oxygen atoms of the ester. The structure T5t prevails in an isolated molecule, and its lowest energy can be explained by the presence of a stronger N-H…O hydrogen bond involving the carbonyl oxygen atom.

A similar explanation can be deduced for the pyrazole with the amine substituent (**2a** and **2b**). For isolated molecules, preference towards the tautomer T5 can be explained by the lack of H…H repulsion between the hydrogen atoms of the amine group and those of the pyrazole ring. The presence of solvent results in the tendency towards the structure T3c for **2a**. However, for **2b** with the ester bond, the increase of environment polarity decreases the gap of energy amongst the structures, which predicts possible equilibrium of the tautomers.

The analysis of the relative differences in energy between the structures for the studied compounds (Appendix A) clearly shows the influence of the environment. In general, the increase of environment polarity lowers the energy of the structure T3t, regardless of the type of substituent. The structure T3t is not stabilized by intramolecular hydrogen bonds; thus, both donor and acceptor groups can be involved in the intermolecular interactions. A similar effect is observed for the structure T5c for the amide compounds **1**–**3a**. In contrast, for the ester compound **1b**, the structures T5c and T5t are stabilized by N-H...O and C-H...O intramolecular hydrogen bonds. Therefore, they are not effectively used in intermolecular interactions, and the relative energies of these structures increase with environment polarity. For the compound **2b**, the environment decreases the relatively energy of the tautomer 3 structures, but for the tautomer 5 structures, the relative energy is maintained, which can be explained by the advantageous interaction of the solvent with the amine substituent. For the compounds with the nitro group, **3a** and **3b**, the tautomer 3 structures are not stabilized much by interactions with the solvent. Even an increase of energy is observed for the structure T3c for **3a**. This can be explained by the presence of N-H...O and C-H...O intramolecular hydrogen bonds with oxygen atoms of the nitro group as acceptors. In contrast, the tautomer 5 structures gain stabilization from both intra-, as well as inter-molecular interactions. Therefore, it can be concluded that both the intra- and inter-molecular interactions play a considerable role. However, the type of substituent determines this influence.

To determine the influence of the substituent on the aromaticity of the pyrazole ring, Nucleus-Independent Chemical Shifts (NICS) were calculated, both for the 3(5)-monosubstituted pyrazole ring (Table 4) and the compounds **1**–**3** (Table 5). As can be seen, the effect is rather small. Therefore, the relatively differences, in particular for the z-component NICS(1)zz obtained 1 Å above the molecular plane, were analyzed, as it turned out to be a more sensitive parameter to the electronic structure [39]. Table 4 shows that the aromaticity of 3(5)-monosubstituted pyrazole compounds decreases as compared to non-substituted pyrazole. Interestingly, regardless of the substituent, this tautomer is preferred, in which the distance between the N-H pyrazole and substituent is longer. The rotation of the amide or ester group has an insignificant influence on aromaticity. Table 5 shows that for all studied compounds **1**–**3**, the tautomer 5 is favored, that is the carbonyl group is closer to the pyrazole N-H group. The tautomer 3 is disfavored in particular for the structure T3c, regardless of the substituent and amide/ester group. It should be noted however that this effect is relatively small for the structures (**1a** and **b**) with the methyl substituent, whereas it is considerable for the structures **2** and **3** with the amino and nitro groups, respectively.

### 2.3. Study in Solution

To determine the type of tautomer in solution, a series of NMR spectra in solvents of various polarities (chloroform, dimethyl sulfoxide, methanol), as well as the NOE experiments were performed (Appendix A). The main differences between the compounds **1**–**3** are presented in Figure 5. For the compounds **1b** and **3b**, the ^1^H NMR spectra show that in each case, only one tautomer is present, regardless of the polarity of the environment studied. However, the NOE experiments show that these tautomers are different. For the compound **1b**, both pyrazole hydrogen atoms (C-H δ_H_ 6.49 and N-H δ_H_ 13.22) show proximity to the methyl substituent (δ_H_ 2.25), which indicates the tautomer 3. The case is similar for the amide analogue **1a**. In contrast, for the compound **3b**, the NOE experiment shows the proximity of both pyrazole hydrogen atoms (C-H δ_H_ 7.53 and N-H δ_H_ 15.24) to the ester methyl group (δ_H_ 3.96). This indicates the tautomer 5. The results are in agreement with X-ray data, as well as theoretical calculations predictions. Interestingly, for the compound **2b** with the amine substituent, a tautomer equilibrium is present in DMSO solution. As can be seen in Table 3, theoretical calculations predicted the smallest differences between the energy of tautomers and conformers for **2b**. Furthermore, the equilibrium is not present in less polar (chloroform) and more polar (methanol) solvents. This shows that the environment has an influence on the choice of tautomer.

Figure 6 shows the ν_S_(C=O)-stretching mode of the Fourier Transform Infrared (FT-IR) spectra for the solutions of the studied molecules **1**–**3b** in weakly-polar chloroform and more polar acetonitrile. In chloroform, the spectra for **1b** and **2b** show single bands (at 1720 and 1725 cm^−1^, respectively), whereas for **3b**, the band shows a small shoulder on the side of higher frequencies, which after deconvolution was resolved on the main band at 1734 cm^−1^ and a much smaller band at 1751 cm^−1^. The calculated stabilities of the structures presented in Table 3 together with the analysis of the theoretical frequencies presented in Table 6 enable assigning for **1b** the band at 1720 cm^−1^ to the structure T3c, for **2b** the band at 1725 cm^−1^ to the structure T5t, and for **3b** the bands at 1734 and 1751 cm^−1^ to the structures T5t and T5c, respectively. In acetonitrile, the spectra for all compounds show two bands; clearly seen for **1b** (at 1730 and 1718 cm^−1^) and after deconvolution for **2b** (at 1731 and 1721 cm^−1^) and **3b** (at 1747 and 1739 cm^−1^). Table 6 shows that these can be assigned to the structures T3c and T3t for **1b** and T5c and T5t for **3b**. In the case of the compound **2b**, all structures are possible. The presented results show that FT-IR spectra show rather conformational changes involving the rotation of the carbonyl substituent and its position in relation to the pyrazole ring.

## 3. Experimental Section

### 3.1. Synthesis

*N-Methyl 5-methyl-1H-pyrazole-3-carboxyamide* (**1a**). Ethyl 5-methyl-1H-pyrazole-3-carboxylate (0.154 g, 1 mM) was dissolved in methanol (1 mL), and methylamine (0.08 mL, 1 mM) was added. The mixture was left for 3 days, and then, the solvents were removed. The product was washed by ethyl acetate (3 × 1 mL); crystallized from a mixture of methanol/diethyl ether. Yield: 40%. ^1^H NMR (400 MHz, DMSO-*d*_6_) δ (ppm): 12.76 (1H, s, N–H_Pz_), 8.04 (1H, s, CON–H), 6.36 (1H, s, C–H_Pz_), 2.72 (3H, s, CH_3_-Pz), 2.23 (3H, s, CH_3_). Melting point: 162–172 °C. TLC CHCl_3_/MeOH (9:1) R_f_ = 0.23.

*Ethyl 5-methyl-1H-pyrazole-3-carboxylate* (**1b**). The compound was purchased from Trimen Chemicals; crystallized from diethyl ether/hexane at about 5 °C. ^1^H NMR (40 0MHz, DMSO-d_6_) δ (ppm): 13.19 (1H, s, N–H_Pz_), 6.49 (1H, s, C–H_Pz_), 4.24 (2H, q, *J* = 7.2, CH_2_), 2.25 (3H, s, CH_3_-Pz), 1.27 (3H, t, *J* = 7.2, CH_3_) (Appendix A). Melting point by DSC: 80 °C (Appendix A). TLC CHCl_3_/MeOH (9:1) R_f_ = 0.44.

*Methyl 5(3)-amino-1H-pyrazole-3(5)-carboxylate (**2b**).* Methyl 3-nitro-1*H*-pyrazole-5-carboxylate (1.71 g, 10 mM) was dissolved in methanol (40 mL), and 10% Pd/C (0.26 g) was added. Hydrogenation was carried out in a Paar apparatus under 5 bar. After 24 h, the catalyst was filtered off and washed by methanol. The solvent was evaporated by reduced pressure, and the residue was crystallized from ethanol (10 mL). Yield: 1.14 g, 81%. Melting point by DSC: 116 and 142 °C (Appendix A). TLC: CHCl_3_/MeOH/AcOH (90:8:2) R_f_ = 0.43. Elemental analysis for C_5_H_7_N_3_O_2_: calculated C, 42.55%; H, 4.96%; N, 29.79%; found: C, 42.70%; H, 5.00%; N, 29.71%. ^1^H NMR (400MHz, DMSO-*d_6_*) δ (ppm): tautomer T5; 12.16 (1H, s, N–H_Pz_), 5.68 (1H, s, C–H_Pz_), 5.19 (2H, s, NH_2_), 3.75 (3H, s, CH_3_), tautomer T3; 12.60 (1H, s, N–H_Pz_), 5.92 (1H, s, C–H_Pz_), 4.81 (2H, s, NH_2_), 3.75 (3H, s, CH_3_) (Appendix A). ^13^C NMR (400 MHz, DMSO-d*_6_*) δ (ppm): tautomer T5; 159.65 (C=O), 148.85 (C-NH_2_), 132.54 (C-C=O), 89.04 (C_4_-H), tautomer T3; 162.87 (C=O), 156.20 (C-NH_2_), 142.35 (C-C=O), 94.27 (C_4_-H).

*Methyl 3-nitro-1H-pyrazole-5-carboxylate* (**3b**). Commercial 3-nitro-1*H*-pyrazole-5-carboxylic acid (99.6% purity, Aldrich) (1.54 g, 10 mM) was dissolved in methanol (15 mL), and 95% H_2_SO_4_ (0.14 mL) was added. The mixture was refluxed for 5 h. The solvent was evaporated under reduced pressure. A precipitate was dissolved in ethyl acetate and washed with 5% NaHCO_3_ (2 × 10 mL) and water (1 × 10 mL). The organic layer was dried with MgSO_4_, and the ethyl acetate was evaporated. The residue was dissolved in dichloromethane and purified on a silica-gel column (60H Merck 1.05553) eluted with DCM:MeOH, from 2%–6% of MeOH; crystallization from a mixture of diethyl ether/hexane. Yield: 1.28 g, 93%. ^1^H NMR (400MHz, DMSO-*d_6_*) δ (ppm): 15.25 (1H, s, N–H_Pz_), 7.54 (1H, s, C–H_Pz_), 3.91 (3H, s, CH_3_) (Appendix A). Melting point by DSC: 148 °C (Appendix A). TLC CHCl_3_/MeOH/AcOH (90:8:2) R_f_ = 0.74.

*Methylammonium 3-nitro-1H-pyrazole-5-carboxylate* (**4**). Methyl 3-nitro-1*H*-pyrazole-5-carboxylate (0.154 g, 1 mM) was dissolved in methanol (1 mL), and methylamine (0.08 mL, 1 mM) was added. After 6 days, the white precipitate was filtered. During washing, the part of the precipitate with methanol dissolved. After several days, salt crystals appeared. Melting point: 195–210 °C.

### 3.2. X-Ray

The single-crystal X-ray diffraction experiments were performed at 100.0(1) K on the Xcalibur diffractometer, equipped with a CCD area detector and a graphite monochromator for the MoKα radiation. The reciprocal space was explored by ω scans with detector positions at a 60-mm distance from the crystal. The diffraction data processing of the studied compounds (Lorentz and polarization corrections were applied) was performed using CrysAlis CCD [40,41]. All structures were solved in the monoclinic crystal system, *P*2_1_/*c, P*2_1_/*n*, and *P*2_1_/*m* space groups, respectively (Table 1), by direct methods and refined by a full-matrix least squares method using the SHELXL14 program [42,43]. The H atoms were located by the difference Fourier synthesis. In both structures, H atoms were refined using a riding model. The structure drawings were prepared using the SHELXTL and Mercury programs [44]. The crystallographic data for the compounds **1**–**5** were deposited at the Cambridge Crystallographic Data Centre as Supplementary Publication Nos. CCDC 1,825,064 (**1**), CCDC 1,825,062 (**2**), CCDC 1,825,063 (**3**), CCDC 1,825,061 (**4**), and CCDC 1,844,066 (**5**). These data can be obtained free of charge via http://www.ccdc.cam.ac.uk/conts/retrieving.html, or from the Cambridge Crystallographic Data Centre, 12 Union Road, Cambridge CB2 1EZ, U.K.; fax: +44 1223 336 033; e-mail: deposit@ccdc.cam.ac.uk.

### 3.3. Computational Procedures

The Gaussian09 package was used to perform the calculations [45]. The M06-2X/6-311++G(d,p) meta-hybrid level of theory [46] and the solvation model [47] were applied. The model compounds **1**–**3** (Figure 1) were analyzed. For each molecule, both tautomers were calculated, in which the carbonyl linkage (ester or amide) was at position 3 or 5. Shortcuts, T3 and T5, were introduced. The flatness of molecules, that is the co-planarity of the pyrazole ring with the ester or amide bond, was assumed. In this case, the torsion angle ψ was close to a value of 0° or 180°. Thus, two possible conformations, cisoidal (T3c, T5c) and transoidal (T3t, T5t), were considered for each tautomer. The ester and amide bonds in the *trans* configuration were calculated.

The isolated molecules (in the gas phase) or in the environment of three solvents (chloroform, acetonitrile, and water) were calculated. Thus, the influence of increased environment polarity was considered. The energy minima were checked by full geometry optimization of the selected structures at the M06-2X/6-311++G (d,p) level of theory in the gas phase, as well as in the solvent-mimicking environment using the solvation model based on the density model. Frequency analyses were carried out to calculate the zero-point vibrational energies and to verify the nature of the minimum state of all the stationary points obtained.

NICS [48,49] was calculated using molecules optimized at the M06-2X/6-311++G (d,p) level of theory. The NICS probes (ghost atoms that are called bq’s in the Gaussian program) were placed at the center of the molecular plane for NICS(0) and 1 Å above the molecular plane for NICS(1) and its z-component NICS(1)zz.

### 3.4. NMR Spectroscopy

A Bruker NMR Spectrometer Ultrashield 400 MHz (2005) together with the Bruker TopSpin Version 1.3 software were applied for data acquisition and processing. The ^1^H and ^13^C NMR spectra were recorded at room temperature in DMSO-*d*_6_ with an internal TMS standard. The NOE difference method (mixing time of 300 ms) within the standard programs was applied to determine the tautomer. The spectra were recorded by using 5000 scans.

### 3.5. FT-IR Spectroscopy

A Nicolet Nexus spectrometer with a deuterated triglycine sulfate detector, flushed with dry nitrogen, was applied. Measurement at room temperature, KBr liquid cell (1.00 mm), concentrations were obtained (3.9–10.9 10^−3^ mol/L), resolution (1 cm^−1^) averaged by using 100 scans. The spectra were obtained by subtraction of the solvent from the sample solutions. GRAMS/A1 Version 9.2 (Thermo Fisher Scientific) was applied to analyze the spectra. Fourier self-deconvolution techniques and the second derivative as an “initial guess” were used to determine the number and position of component bands. The quantum mechanical calculations were also taken into account to determine the number of component bands. The curve-fitting procedure with a mixed (Gauss–Lorentz) profile was applied to determine the accurate band position.

## 4. Conclusions

The general concept, which arose from the presented studies, indicated that the tendency to adopt a given tautomer is connected with the substituents at the pyrazole ring. The preferred tautomer has nitrogen with a lone pair closer to a more electron-withdrawing substituent. When the methyl group was present, the lone pair at the pyrazole ring was closer to the ester/amide group. Thus, the tautomer 3 was preferred. When the nitro group was present, the lone pair at the pyrazole ring was further from the ester/amide group. Thus, the tautomer 5 was preferred. The same was the case of 3-nitro-*1H*-pyrazole-5-carboxylate, which revealed tautomer 5 in the crystal state. The analysis of the compounds in the Cambridge Crystallographic Data, which also have phenyl, nitrile, carboxy, hydroxyl, and fluoroalkyl substituents, further confirmed this tendency. The theoretical methods showed that at least three factors had an influence on the annular tautomerism of pyrazole substituted by the ester/amide group: the pyrazole ring aromaticity determined by the second substituent, internal hydrogen bonds, and intermolecular interactions, including the polarity of the environment. The preferences towards the specific tautomer and conformation seem to depend on a balance between these factors. The influence of these factors was depicted by the compound with the amino substituent. Theoretical calculations on isolated molecules predicted the preference of tautomer 5. However, the increase of the environment polarity diminished the gap in energy; thus, the NMR spectrum in DMSO solution showed the equilibrium of tautomers. In the crystal state, the tautomer 3 was present, as determined by the X-ray method. The FTI-IR analysis also showed equilibria, but they were connected rather with the conformational arrangement of the ester/amide carbonyl group and pyrazole ring.

## Figures and Tables

**Figure 1 molecules-24-02632-f001:**
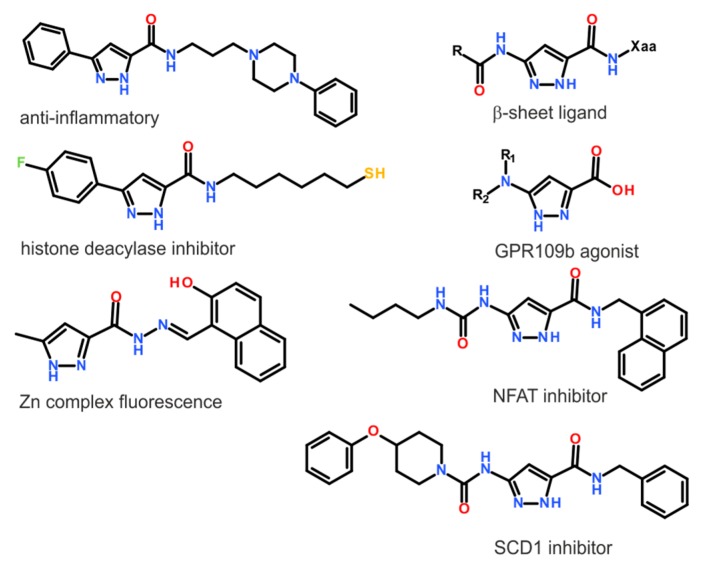
Examples of 1H-pyrazole compounds with the 3(5)-carbonyl group.

**Figure 2 molecules-24-02632-f002:**
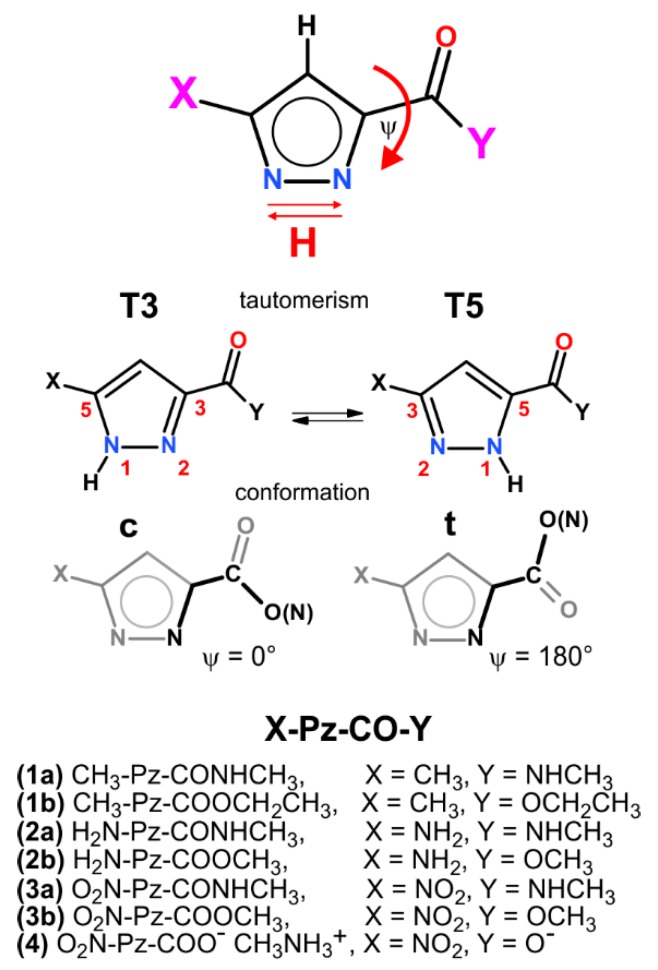
Schematic representation of tautomers in 3(5)-1H-pyrazoles, conformation of the 3(5)-carbonyl group, and numeration of the series of studied compounds.

**Figure 3 molecules-24-02632-f003:**
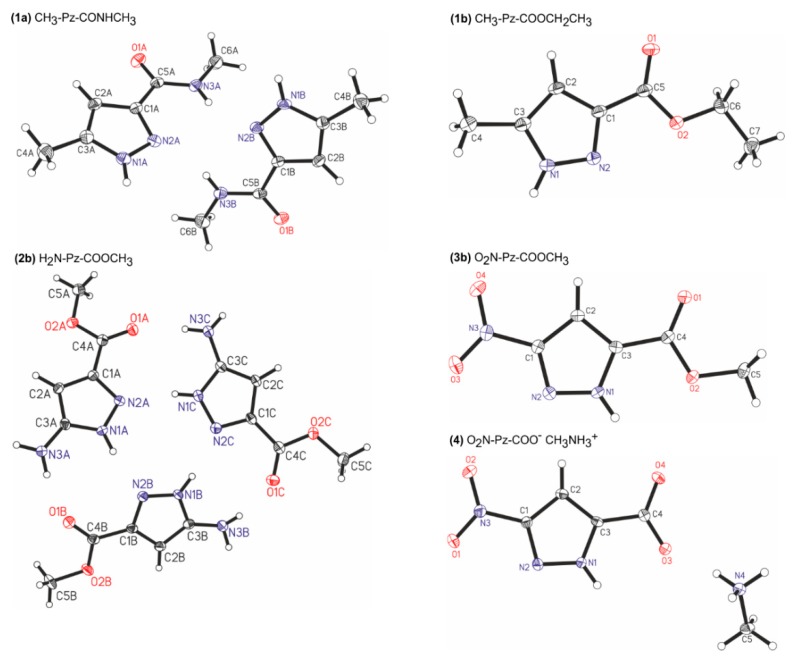
Molecular structures of the studied molecules in the crystal structure in the asymmetric part of the unit cell. Displacement ellipsoids are drawn at the 50% probability level.

**Figure 4 molecules-24-02632-f004:**
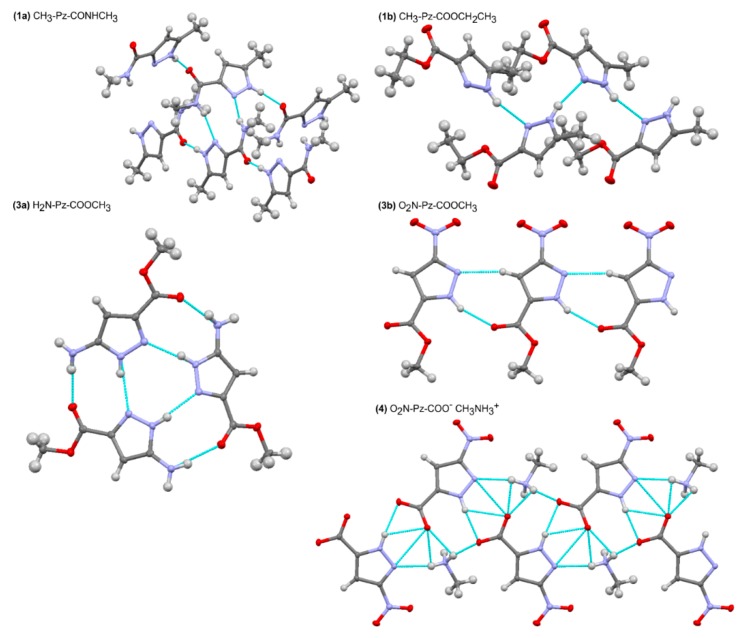
Molecular interactions of the studied molecules with visualization of the hydrogen bonds.

**Figure 5 molecules-24-02632-f005:**
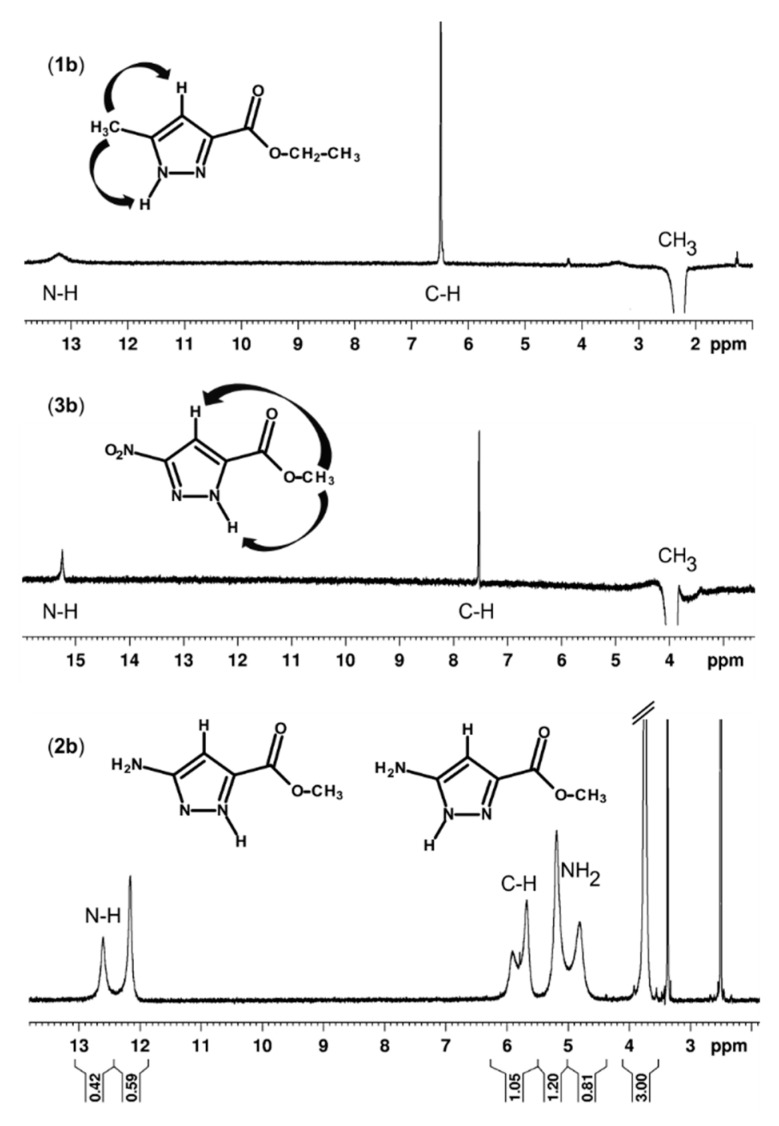
NMR-NOE spectra in DMSO-d_6_ for the compounds **1b** and **3b**, which show the preferred tautomer and ^1^H spectrum for **2b**, which shows the equilibrium of the tautomers.

**Figure 6 molecules-24-02632-f006:**
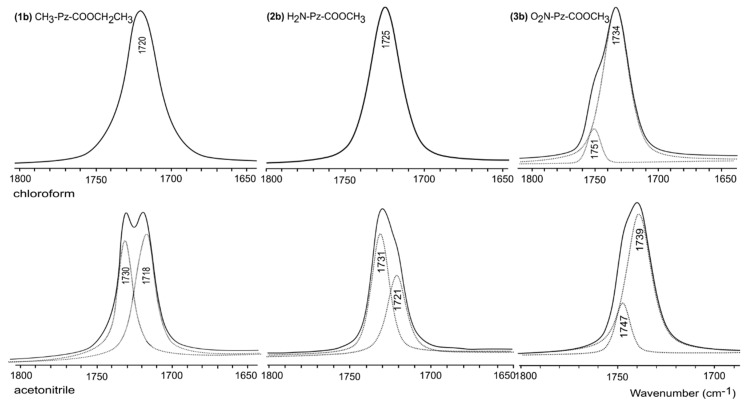
Selected FT-IR spectra of the studied molecules **1**–**3b** in the ν_S_(C=O) region.

**Table 1 molecules-24-02632-t001:** Crystal data and structure refining for the studied compounds.

	*1a*	*1b*	*2b*	*3b*	*4*
Chemical formula	C_6_H_9_N_3_O	C_7_H_10_N_2_O_2_	C_5_H_7_N_3_O_2_	C_5_H_5_N_3_O_4_	C_5_H_8_N_4_O_4_
*M* _r_	139.16	154.17	141.14	171.12	188.15
Crystal system, space group	Monoclinic, *P*2_1_/*c*	Monoclinic, *P*2_1_/*n*	Monoclinic, *P*2_1_/*n*	Monoclinic, *P*2_1_/*m*	Monoclinic, *P*2_1_/*n*
*a*, *b*, *c* (Å)	14.7736 (4), 7.2104 (2), 13.6295 (4)	9.7186 (7), 6.0828 (4), 13.4976 (11)	10.8040 (5), 13.6444 (5), 13.8883 (6)	5.5232 (3), 6.1484 (3), 10.8318 (5)	3.7803 (1), 22.9836 (7), 9.0425 (3)
β (°)	90.800 (3)	102.810 (7)	110.173 (5)	102.679 (5)	98.082 (3)
*V* (Å^3^)	1451.72 (7)	778.07 (10)	1921.74 (15)	358.87 (3)	777.85 (4)
*Z*	8	4	12	2	4
µ (mm^−1^)	0.09	0.1	0.12	0.14	0.14
Crystal size (mm)	0.5 × 0.4 × 0.2	0.5 × 0.3 × 0.1	0.4 × 0.3 × 0.2	0.5 × 0.4 × 0.2	0.4 × 0.3 × 0.2
No. of measured, independent and observed [*I* > 2σ(*I*)] reflections	9524, 2826, 2196	5041, 1523, 1130	9264, 3742, 2660	2439, 776, 685	5234, 1530, 1186
*R* _int_	0.024	0.037	0.027	0.013	0.027
(sin θ/λ)_max_ (Å^−1^)	0.617	0.616	0.617	0.617	0.617
Refinement					
*R*[*F*^2^ > 2σ(*F*^2^)], *wR*(*F*^2^), *S*	0.037, 0.103, 1.08	0.038, 0.103, 0.88	0.034, 0.085, 0.77	0.027, 0.076, 1.11	0.029, 0.070, 0.95
No. of reflections	2826	1523	3742	776	1530
No. of parameters	185	102	292	79	120
H-atom treatment	H-atom parameters constrained	H-atom parameters constrained	H atoms treated by a mixture of independent and constrained refinement	H atoms treated by a mixture of independent and constrained refinement	H-atom parameters constrained
Δρ_max_, Δρ_min_ (e Å^−3^)	0.26, −0.26	0.20, −0.27	0.17, −0.25	0.29, −0.19	0.24, −0.24

**Table 2 molecules-24-02632-t002:** Tautomers/conformers of the *1H*-pyrazole residue (-Pz-CO-) with carbonyl at position 3(5) found in the solid state.

X-Pz-CO-	Tautomer (T3/5)/Conformer (c/t)			
X-	T3c	T3t	T5c	T5t
Me-	7			
H-	2	3		
Ph-	12	1	3	2
OOC-Pz-COOH	11	5	1	2
NC-			1	3
-CF-			6	2
HO-				1

**Table 3 molecules-24-02632-t003:** Local minima and the selected thermodynamic properties of the studied compounds **1–3** with reference both to tautomers and the rotation of the carbonyl group.

	*In Vacuo*			*Chloroform*			*Acetonitrile*			*Water*		
	ψ	Energy	ΔE	ψ	Energy	ΔE	ψ	Energy	ΔE	ψ	Energy	ΔE
Conformer Code↓	(°)	(Ha)	(kcal/mol)	(°)	(Ha)	(kcal/mol)	(°)	(Ha)	(kcal/mol)	(°)	(Ha)	(kcal/mol)
(**1a**) CH_3_-Pz-CONHCH_3_												
T3c	0.0	−473.322883	**0.00**	0.0	−473.340504	**0.00**	0.0	−473.344429	**0.00**	0.0	−473.344742	**0.00**
T3t	180.0	−473.312263	6.66	180.0	−473.334178	3.97	180.0	−473.340706	2.34	180.0	−473.342131	1.64
T5c	24.4	−473.314249	5.42	24.1	−473.333208	4.58	17.6	−473.338227	3.89	18.4	−473.339003	3.60
T5t	180.0	−473.321441	0.90	180.0	−473.337598	1.82	180.0	−473.341534	1.82	180.0	−473.341632	1.95
(**1b**) CH_3_-Pz-COOCH_2_CH_3_												
T3c	0.0	−532.474789	1.83	0.0	−532.492603	**0.00**	0.0	−532.496648	**0.00**	0.0	−532.494392	**0.00**
T3t	180.0	−532.473347	2.73	180.0	−532.491806	0.50	180.0	−532.496280	0.23	180.0	−532.494119	0.17
T5c	0.2	−532.477188	0.32	0.4	−532.491509	0.69	0.3	−532.494031	1.64	0.0	−532.490849	2.22
T5t	180.0	−532.477703	**0.00**	180.0	−532.491767	0.52	180.0	−532.494172	1.55	180.0	−532.490747	2.29
(**2a**) H_2_N-Pz-CONHCH_3_												
T3c	0.0	−489.374774	1.25	0.0	−489.397950	**0.00**	0.0	−489.404669	**0.00**	0.0	−489.405337	**0.00**
T3t	−179.9	−489.365033	7.36	−180.0	−489.392782	3.24	−180.0	−489.402127	1.59	−180.0	−489.404202	0.71
T5c	24.2	−489.371558	3.26	25.3	−489.393897	2.54	18.1	−489.400630	2.53	5.1	−489.401874	2.17
T5t	−180.0	−489.376759	**0.00**	180.0	−489.397510	0.28	180.0	−489.403626	0.65	−180.0	−489.404197	0.72
(**2b**) H_2_N-Pz-COOCH_3_												
T3c	0.0	−509.246751	4.67	0.0	−509.267576	1.80	0.0	−509.274622	0.43	0.0	−509.273901	0.27
T3t	180.0	−509.245885	5.21	180.0	−509.267586	1.80	180.0	−509.275108	0.13	180.0	−509.274337	**0.00**
T5c	0.0	−509.253108	0.68	0.4	−509.270059	0.25	0.4	−509.275012	0.19	0.2	−509.273864	0.30
T5t	180.0	−509.254195	**0.00**	179.9	−509.270450	**0.00**	179.8	−509.275315	**0.00**	−179.9	−509.273994	0.22
(**3a**) O_2_N-Pz-CONHCH_3_												
T3c	0.0	−638.517709	0.63	0.0	−638.535327	2.29	0.0	−638.539811	3.59	0.0	−638.538274	3.81
T3t	−155.5	−638.508452	6.43	−174.4	−638.529928	5.68	−174.3	−638.536869	5.44	−170.8	−638.536671	4.82
T5c	26.8	−638.510067	5.42	24.8	−638.533577	3.39	22.6	−638.541327	2.64	20.4	−638.540059	2.69
T5t	180.0	−638.518707	**0.00**	−177.3	−638.538973	**0.00**	180.0	−638.545531	**0.00**	−179.9	−638.544350	**0.00**
(**3b**) O_2_N-Pz-COOCH_3_												
T3c	0.0	−658.389922	2.95	0.0	−658.404657	3.63	0.0	−658.409316	3.72	0.0	−658.405911	3.65
T3t	180.0	−658.389303	3.34	180.0	−658.404464	3.75	180.0	−658.409578	3.56	180.0	−658.406313	3.40
T5c	0.0	−658.393160	0.92	0.0	−658.409513	0.59	0.0	−658.414409	0.53	0.0	−658.411099	0.39
T5t	180.0	−658.394621	**0.00**	180.0	−658.410447	**0.00**	180.0	−658.415251	**0.00**	180.0	−658.411725	**0.00**

**Table 4 molecules-24-02632-t004:** NICS parameters for 3(5)-monosubstituted pyrazole.

Compound	NICS(0)	NICS(1)	NICS(1)zz
Pyrazole	−13.51	−11.37	−33.87
3-Methyl-1H-pyrazole	−12.57	−13.22	−31.38
5-Methyl-1H-pyrazole	−12.13	−12.12	−30.52
1H-pyrazol-3-amine (3-Aminopyrazole)	−11.97	−8.93	−26.42
1H-pyrazol-5-amine (5-Aminopyrazole)	−11.94	−8.72	−25.71
3-Nitro-1H-pyrazole	−13.50	−10.63	−29.74
5-Nitro-1H-pyrazole	−12.43	−10.36	−28.76
N-Methyl-1H-pyrazole-3-carboxamide (ψ = 0°)	−12.81	−10.89	−30.98
N-Methyl-1H-pyrazole-3-carboxamide (ψ = 180°)	−12.95	−10.99	−31.00
N-Methyl-1H-pyrazole-5-carboxamide (ψ = 0°)	−12.61	−10.90	−32.07
N-Methyl-1H-pyrazole-5-carboxamide (ψ = 180°)	−12.28	−10.86	−30.69
Methyl 1H-pyrazole-3-carboxylate (ψ = 0°)	−12.88	−10.98	−31.14
Methyl 1H-pyrazole-3-carboxylate (ψ = 180°)	−12.88	−11.03	−31.16
Methyl 1H-pyrazole-5-carboxylate (ψ = 0°)	−12.29	−10.89	−30.85
Methyl 1H-pyrazole-5-carboxylate (ψ = 180°)	−12.21	−10.88	−30.68

**Table 5 molecules-24-02632-t005:** NICS parameters for the studied compounds **1**–**3**.

Code	NICS(0)	NICS(1)	NICS(1)zz	∆NICS(1)zz
(**1a**) CH_3_-Pz-CONHCH_3_				
**T3c**	−11.57	−10.08	−27.86	0.00
**T3t**	−11.68	−10.12	−27.88	−0.02
**T5c**	−11.57	−10.05	−27.94	−0.08
**T5t**	−11.35	−10.23	−28.09	−0.23
(**1b**) CH_3_-Pz-COOCH_2_CH_3_				
**T3c**	−11.61	−10.06	−27.93	0.00
**T3t**	−11.65	−10.19	−28.02	−0.08
**T5c**	−11.37	−10.76	−28.3	−0.36
**T5t**	−11.33	−10.29	−28.12	−0.18
(**2a**) H_2_N-Pz-CONHCH_3_				
**T3c**	−11.48	-8.45	-23.36	0.00
**T3t**	−11.64	-8.56	-23.44	-0.08
**T5c**	−11.45	-8.95	-25.81	-2.44
**T5t**	−11.09	-8.75	-24.01	-0.65
(**2b**) H_2_N-Pz-COOCH_3_				
**T3c**	−11.70	−8.61	−23.60	0.00
**T3t**	−11.72	−8.63	−23.71	−0.11
**T5c**	−11.43	−9.08	−24.84	−1.24
**T5t**	−11.40	−9.10	−24.86	−1.26
(**3a**) O_2_N-Pz-CONHCH_3_				
**T3c**	−12.12	−10.00	−26.33	0.00
**T3t**	−12.50	−10.28	−27.40	−1.07
**T5c**	−13.06	−10.29	−27.99	−1.66
**T5t**	−12.72	−10.33	−27.16	−0.82
(**3b**) O_2_N-Pz-COOCH_3_				
**T3c**	−12.30	−10.17	−26.60	0.00
**T3t**	−12.36	−10.22	−26.72	−0.12
**T5c**	−12.77	−10.33	−27.28	−0.68
**T5t**	−12.76	−10.35	−27.22	−0.62

**Table 6 molecules-24-02632-t006:** Calculated and measured frequencies of the studied molecules **1**–**3b** in the ν_S_(C=O) region in chloroform and acetonitrile.

Code	Chloroform			Acetonitrile		
	Calc.	Scaled ^a^	Measured	Calc.	Scaled ^b^	Measured
(**1b**) CH_3_-Pz-COOCH_2_CH_3_						
T3c	1803.76	1715	1720	1799.13	1716	1718
T3t	1824.47	1735		1814.27	1731	1730
(**2b**) H_2_N-Pz-COOCH_3_						
T3c	1807.09	1718		1801.77	1719	1721
T3t	1826.99	1737		1818.40	1735	1731
T5c	1827.74	1737		1820.13	1736	1731
T5t	1808.87	1720	1725	1805.58	1723	1721
(**3b**) O_2_N-Pz-COOCH_3_						
T5c	1840.61	1750	1751	1831.87	1748	1747
T5t	1824.01	1734	1734	1822.54	1739	1739

^a^ scaling factor 0.951. ^b^ scaling factor 0.954.

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
