# Peer review of "Annular Tautomerism of 3(5)-Disubstituted-1H-pyrazoles with Ester and Amide Groups"

_molecules, 2019, doi:10.3390/molecules24142632_

Round 1

Reviewer 1 Report

The manuscript describes an interesting exploration of prototropic annular tautomerism in derivatives of pyrazole-3(5)-carboxylic acid. Overall, authors's conclusions are well supported by experimental and theoretical data.  These data well agree with the earlier reported for similar structures, they further contribute to the field and will attract attention of the Molecules readership.  

There are some English problems and issues with terminology and data presentation.

Line 76: Co-planarity is essential for the pi-electron conjugation, but it is not a sufficient confirmation of such conjugation.  The conjugation is obviously takes place, but since the C1-C5 bond length (Table S1) is even longer than typical single bond between two sp2 hybridized carbon atoms, the role of this conjugation should not be overestimated. 

Lines 91-92: "three independent molecules are present in asymmetric unit" would be more accurate. 

Line 96: See comment for line 76 applied to C1-C4 bonds of 2b.

Section 2.2 contains particularly many problems with English from incomplete sentences (line 164) to grammatically incorrect constructions and randomly inserted commas. Besides, there are issues with methodology and terminology.  In line 176, authors use coplanarity observed in theoretical calculations as an evidence of conjugation.  However, in line 325 explaining the methodology of their theoretical calculations they postulate that this coplanarity was assumed. Authors use term "internal hydrogen bonds" (lines 173, 174, 194, 198, and 206). This term is unclear and mistakenly opposed to intramolecular interactions (lines 198 and 206).  Please correct. 

Line 231: Since only data for DMSO solution are discussed in this sentence, so it is unclear that authors mean by "regardless of the polarity of environment studied." 

Lines 275-276: Please report define multiplets (t and q) as a single value instead of intervals. Please check and correct their J values too as they are incorrect for these two signals.

Please be consistent in you abbreviations for DMSO.

Please correct spellings of "difluorovinyl" and "ferrocenyl" in column 4 of Table 7S. 

After addressing these comments and some English editing this paper will be publishable in Molecules. 

Author Response

Dear Sirs,

Thank you very much for the constructive and helpful review. Also, I would like to thank for the time and attention you spent to improve the submitted work.

Please, find enclosed the corrected version of the manuscript entitled: Annular tautomerism of 3(5)-disubstituted-1H-pyrazoles with ester and amide group (ID: molecules-556257). All comments from Reviewers have been included in the revised version.  The changes are marked blue in the text.

Therefore, we would be very grateful, if you would consider our work in Molecules.

The detailed respond is given below:

Reviewer 1

We improved English language of the manuscript, in particular, connected with terminology. We agree that co-planarity of the pyrazole ring and ester/amide group does not necessarily have to be associated with the presence of considerable pi-electron conjugation. Therefore, such statements was basically removed in the new version. Thank you for more accurate description of the crystal state of compound 2b. It was incorporated in the text. Indeed, section 2.2. contained many English errors. It was improved. Also, the term of intra- inter-molecular interactions was corrected. The phrases in the section 2.2. was shortened to make the idea more clear. According to remaining remarks, DMSO abbreviation was unified, coupling constant in the NMR data was corrected, some spelling in Supplementary Materials was changed. Once again, thank you very much for your help.

Yours faithfully,

Dawid Siodłak

Reviewer 2 Report

The manuscript of Siodłak et al. is a very interesting contribution to the study of the annular tautomerism of 3(5)-disubstituted-1H-pyrazoles with ester and amide groups.

The methodology is appropriated (X-ray, solution Nmr and theoretical calculations), the results are scientifically sounded and their presentation is excellent.

A minor point is that in the Abstract is necessary to clarify the following statement: “The X-ray experiment shows in crystal state tautomer 3 for the compounds with methyl (1a, 1b) and amino 11 (2b) group, whereas for nitro group (3b, 4) the tautomer 5”.

I would suggest to change it by: “The X-ray experiment shows in crystal state tautomer bearing the ester or amide groups in position 3 (tautomer 3) for the compounds with methyl (1a, 1b) and amino 11 (2b) group, whereas for nitro group (3b, 4) they are in position 5 (tautomer 5)”

Similarly in the Conclusions.

In the Experimental, the authors need to revise the assignments of the 1H NMR and 13C NMR data, some of them are wrong. According to the integration of the signals in the 1H spectrum of 2b depicted  in Figure 5, the signals corresponding to the minor tautomer (40%)  appear to be 12.60, 5.92 and 4.81 ppm, and those of the major tautomer  (60%) 12.16, 5.68, 5.19 ppm. In what concerns  the 13C NMR chemical shifts given, the signals at 89.04 and  94.27 ppm are the ones of the C4-H of the pyrazole ring (see references  35 and 36 of the manuscript between others), and not those of C5. The  authors need to perform additional NMR experiments  to assign the remaining carbons to major and minor tautomers; or just  state the chemical shifts with no assignments.

Finally, if the authors continue their research on tautomerism it will be worth to incorporate the solid-state NMR spectroscopic data.

Author Response

Dear Sirs,

Thank you very much for the constructive and helpful review. Also, I would like to thank for the time and attention you spent to improve the submitted work.

Please, find enclosed the corrected version of the manuscript entitled: Annular tautomerism of 3(5)-disubstituted-1H-pyrazoles with ester and amide group (ID: molecules-556257). All comments from Reviewers have been included in the revised version. The changes are marked blue in the text.

Therefore, we would be very grateful, if you would consider our work in Molecules.

The detailed respond is given below:

Reviewer 2

The statement in the Abstract concerning the X-ray measurement was changed according advice.

The assignment of the NMR data of the compound 2b was corrected. The 13C NMR spectra was added to Supplementary Materials (Figure 15S). In addition, we decided to place the DSC data for the compounds 1-3b (Figure 22-24S), because it can be interesting for readers .

Yours faithfully,

Dawid Siodłak